# Measurable Residual Disease and Clonal Evolution in Acute Myeloid Leukemia from Diagnosis to Post-Transplant Follow-Up: The Role of Next-Generation Sequencing

**DOI:** 10.3390/biomedicines11020359

**Published:** 2023-01-26

**Authors:** Alessandra Sperotto, Maria Teresa Bochicchio, Giorgia Simonetti, Francesco Buccisano, Jacopo Peccatori, Simona Piemontese, Elisabetta Calistri, Giulia Ciotti, Elisabetta Pierdomenico, Roberta De Marchi, Fabio Ciceri, Michele Gottardi

**Affiliations:** 1Onco Hematology, Department of Oncology, Veneto Institute of Oncology, IOV-IRCCS, 31033 Padua, Italy; 2Biosciences Laboratory, IRCCS Istituto Romagnolo per lo Studio dei Tumori (IRST) “Dino Amadori”, 46046 Meldola, Italy; 3Department of Biomedicine and Prevention, University of Rome Tor Vergata, 00133 Rome, Italy; 4Hematology and Hematopoietic Stem Cell Transplantation Unit, IRCCS San Raffaele Scientific Institute, 20132 Milan, Italy

**Keywords:** acute myeloid leukemia, measurable residual disease, clonal evolution, next-generation sequencing, allogeneic stem cell transplantation

## Abstract

It has now been ascertained that acute myeloid leukemias—as in most type of cancers—are mixtures of various subclones, evolving by acquiring additional somatic mutations over the course of the disease. The complexity of leukemia clone architecture and the phenotypic and/or genotypic drifts that can occur during treatment explain why more than 50% of patients—in hematological remission—could relapse. Moreover, the complexity and heterogeneity of clone architecture represent a hindrance for monitoring measurable residual disease, as not all minimal residual disease monitoring methods are able to detect genetic mutations arising during treatment. Unlike with chemotherapy, which imparts a relatively short duration of selective pressure on acute myeloid leukemia clonal architecture, the immunological effect related to allogeneic hematopoietic stem cell transplant is prolonged over time and must be overcome for relapse to occur. This means that not all molecular abnormalities detected after transplant always imply inevitable relapse. Therefore, transplant represents a critical setting where a measurable residual disease-based strategy, performed during post-transplant follow-up by highly sensitive methods such as next-generation sequencing, could optimize and improve treatment outcome. The purpose of our review is to provide an overview of the role of next-generation sequencing in monitoring both measurable residual disease and clonal evolution in acute myeloid leukemia patients during the entire course of the disease, with special focus on the transplant phase.

## 1. Introduction

Acute myeloid leukemias (AMLs) are a heterogeneous group of clonal disorders occurring after chromosomal and or genomic alterations in hematopoietic stem cells or progenitor cells [1,2,3,4,5].

Although 70% of patients with AMLs attain morphologic complete remission (mCR) with intensive induction chemotherapy (IIC), approximately 50% of these patients experience relapse [6,7]. Moreover, disease recurrence represents the primary reason for treatment failure and death also after allogeneic hematopoietic stem cell transplant (alloHSCT) [8].

Therefore, a significant effort is needed to translate the advances in understanding the genetic complexity of AMLs into routine clinical practice.

Unlike other disease that are characterized by a single somatic mutation (as in chronic myeloid leukemia with Abelson murine leukemia gene translocation onto the breakpoint cluster region (BCR/ABL translocation) or in acute lymphoblastic leukemia with T-cell receptor (TCR) rearrangements), individual AMLs are potentially composed of clones and subclones that harbor multiple mutations, making each patient genetically unique [9,10,11,12,13].

Traditionally, in AMLs, molecular and cytogenetic risk classification has been used to predict disease progression and then survival, allowing patient selection for more intensive treatments, including transplant [14,15,16,17]. However, conventional cytogenetic analysis is not enough for understanding the underlying disease clonal architecture and the genotypic/phenotypic drifts that can occur during the treatment course [12].

One of the most important advances in this setting was the introduction of next-generation sequencing (NGS), which enables reliable detection of patient-specific mutations (and other alterations) including complete gene assessment at diagnosis, complete remission (CR), and relapse [18].

Despite the fact that the relevance of NGS at diagnosis has now been ascertained, its role in other phase of the disease—during treatment, at relapse, before and after transplant—is less clear.

In our review, we try to provide an overview of the clonal evolution of AML monitoring by NGS during the entire course of the disease, including in the alloHSCT setting.

## 2. Acute Myeloid Leukemia

Acute myeloid leukemia is defined by the malignant clonal expansion of a progenitor cell coupled to a differentiation block [1,2,3,4,5]. Among adults, the incidence of AML had been higher than that of other leukemias such as chronic myeloid and acute lymphoblastic leukemia; after 2017, it was surpassed by chronic lymphocytic leukemia [19,20,21].

AMLs account for the highest percentage (more than 60%) of leukemic deaths and the worst outcome (5-year survival around 24%), with respect to other leukemias [21].

The age-adjusted incidence of AML is 4.3 per 100,000 annually in the United States, with a median age at diagnosis of 68 years; incidence increases according to age.

The etiology is heterogeneous: prior exposure to chemo- and/or radiotherapy and occupational or environmental DNA-damaging agents, among others. However, most cases of AMLs remain without a clear etiology.

Curative therapies, including intensive chemotherapy and alloHSCT, are generally applicable to a minority of patients (younger and fitter), while most older individuals exhibit poor prognosis and survival due to less intensive programs being employed.

Differences in patients’ outcomes are influenced by disease molecular and cytogenetic characteristics, age, treatment related complications, and concomitant patient comorbidities.

After many years without therapeutic advances, several new molecules have been approved for treatment and are expected to impact patient’s outcomes, especially for older patients and those with refractory disease.

## 3. Next-Generation Sequencing Platforms as Used in Hematology

NGS is a method of DNA and RNA sequencing. Millions of DNA fragments can be sequenced simultaneously; the analysis may be limited to selected segments of certain genes or to the entire exome or genome [22]. In the beginning, NGS platforms primarily were used for research purposes, but they are increasingly being integrated as irreplaceable diagnostic tools into routine clinical practice.

In the hematological setting, with NGS we can detect at least one driver mutation in more than 90% of AML patients and at least two in more than 85%, allowing us to uncover leukemias that lack classical chromosomal and or gene alterations [23].

NGS, because it investigates the genetic profile several layers deeper than conventional techniques, introduces a new concept of disease understanding, opening up opportunities for new therapeutic options.

Of particular interest is the identification, by NGS, of mutations that can predict outcome (early relapse, progression, or resistant disease); detection of the co-occurrence of newly discovered and driver mutations offers a new concept regarding prognosis.

There are several commercially available NGS myeloid panels: Illumina (Illumina TruSight Myeloid panel); Qiagen (Human Myeloid Neoplasms QiaSeq DNA Panel); ThermoFisher (Oncomine Myeloid panel); Archer (FusionPlex panel); SOPHiA GENETICS (Myeloid, Myeloid Plus or Extended Myeloid Solution) [24].

These panels cover between 25 and more than 98 relevant genes directly or indirectly involved in the pathophysiology of myeloid neoplasms (transcription factors, epigenetic modifiers, signal molecules, tumor suppressors, and cohesion-complex gene), as entire genes or mutational hotspots.

Now, with Illumina and Archer RNA-sequencing panels, we can simultaneously detect—in a single assay—more than 300 fusion transcripts, including previously unknown partner genes, with potential new prognostic and/or therapeutic value [25].

Moreover, by using dedicated pipelines, RNA-sequencing panels can lead to the identification of coding region variants.

Compared to first-generation sequencing (Sanger), which has a limit of detection of approximately 20% variant allele frequency (VAF), NGS-targeted resequencing increases the sensitivity from around 1% to 5%, depending on allele overlay [26,27]. With the novel NGS-based applications that are constantly emerging, even low-frequency AML subclones can be detected.

Limits to the NGS sensitivity are related to the VAF of a mutated gene in respect to the normal one; for VAFs falling into the lower range (e.g., 0.1%), the depth of cover for considering a variant as true needs to be high (10,000×), and it must overcome the intrinsic error of the amplification reaction inherent in NGS [28].

Error-corrected or barcoded sequencing (ECS) is one of several methods developed for optimizing NGS sensitivity (for example, by the incorporation of a unique molecular index (UMI) to each molecule captured), enabling the detection of clones as rare as 1:10,000 [29,30]. Another method—also based on molecular barcoding—comprises single molecular inversion probes (smMIPs); this is a technology based on multiplexed targeted sequencing and error correction schemes [31,32]. These innovative techniques, circumventing standard NGS sensitivity, hold great promise [33].

The major challenges with NGS are cost, the time required for diagnostic assay, the technical aspects of adapting the methods to clinical settings, and the clinical interpretation of the data during treatment.

The sequencing of patient samples in large retrospective studies is currently drawing associations between NGS data and patient outcomes, with some of the associations already finding clinical applicability in prospective clinical trials [34,35].

## 4. Measurable Residual Disease: Next-Generation Sequencing Vs. Conventional Techniques

MRD assessment in AMLs is crucial for several reasons: for detecting disease remission as deeply as possible with the available methods, for defining the risk of relapse in patients in remission, and for the prompt detection of impending relapse, thus enabling early intervention [36].

It is therefore reasonable to assume that more than one method of MRD detection should be employed in order to detect a very low disease burden, reduce false positive results due to competing phenomena (e.g., clonal hematopoiesis of indeterminate potential (CHIP)), reveal changes in clone architecture during treatment, and account for the set of possible disease-characterizing markers that may develop with relapse [12,37,38,39,40].

For clinical purposes in the AML setting, to be efficacious, the MRD method should detect one abnormal cell in 10,000, with a sensitivity on the order of 10^4^.

The two most extensively employed methodologies for MRD detection (multiparameter flow cytometry MRD (MFC-MRD) and molecular MRD (Mol-MRD) assessed by quantitative Polymerase Chain Reaction (qPCR) can meet this threshold [41,42].

Various studies and a systematic meta-analysis of 81 publications have shown the prognostic value of MRD detection (with conventional methodologies: MFC-MRD and Mol-MRD) for relapse and overall survival (OS); both with intensive and more recently less-intensive treatments [43,44,45,46,47,48].

The sensitivity in the order of 10^4^ explains, on one hand, why relapse still occurs, also within MRD-negative patients and, on the other hand, why not all MRD-positive patients will relapse. In the first case, this is because pathological subclones can remain undetected by standard MRD techniques, and so a negative MRD may not describe a complete disease eradication, but only a disease below the MRD threshold.

In the second case, MRD values could remain detectable at a low level without prognostic significance and therefore be considered below the threshold linked to prognosis, and then be operationally called negative [43].

Clonal heterogeneity and dynamic clonal evolution of AMLs from diagnosis to relapse are two other reasons that explain—in part—why MRD negative patients could relapse. Clonal heterogeneity and evolution have been demonstrated in longitudinal studies on the same patient; analyzing bone marrow samples from diagnosis to relapse of a heterogeneous pattern of relapse has been shown [49,50].

Therefore, in a dynamic disease such as AML, where relapse can occur because of a subclone with a different genetic profile than the original one, a “dynamic” method for MRD assessment is required, in addition to conventional ones, that reveals some limits. In fact, Mol-MRD depends on amplification of gene-specific primers with the limits of type of gene mutation and presence of alternative splice sites that could influence suboptimal gene amplification [39]. On the other hand, MFC-MRD describes a leukemia associated or different from normal phenotypes, identified by 3 to 10 fluorescent markers [38].

While emerging exploratory technologies, such as NGS, can assay for the multiple genetic markers associated with AML and is also able to detect novel mutations appearing within the exon sequence, it could also be considered a “dynamic” MRD test, capable of detecting the dynamic clonal evolution of AML during treatment [11].

When an NGS-MRD-based strategy is utilized, certain strategies should be employed: error-corrected approaches should be preferred; germline mutations must be recognized and excluded; and mutations related to pre-malignant clonal hematopoiesis (e.g., DNA methyltransferase, DNMT3A; DNA demethylase, TET2; putative polycomb group of protein, ASXL1, the so called DTA mutations) should not be included as MRD probes [43,51,52,53].

Despite the fact that, previously, NGS as MRD assessment has not yet been standardized, several studies have demonstrated a similar sensitivity and specificity of NGS compared to standard mol-MRD.

Based on mutation detection in nucleophosmin1 (NPM1) and internal tandem duplication of FMS-like tyrosine kinase 3 (FLT3-ITD) genes, Thol et al. [54] showed that NGS had assured MRD assessment and 95% concordance with mol-MRD (RT-qPCR) for mutated NPM1. Data confirmed by Spencer et al. in 51 AML patients, with a 100% sensitivity (95% CI = 83% to 100%) and 100% specificity (95% CI = 88% to 100%), NGS analysis provided accurate ITD insertion sizes and was able to detect the ITD alleles present at estimated frequencies as low as 1% [55].

Although more expensive and laborious than conventional MRD, NGS allows the tracking of distinct mutational patterns and enables the discrimination from normal hematopoiesis with greater accuracy, especially among those patients with rare gene mutations (Figure 1).

As described above, the pre-therapy baseline sample and turnaround time required for the single diagnostic assay still represent limits for NGS employed as MRD test.

## 5. Relevant Timepoints for Next-Generation Sequencing Measurable Residual Disease Monitoring

### 5.1. Chemotherapy Setting: Induction—Consolidation Phase

Employing NGS for MRD assessment in AML patients has been proposed since the very beginning of NGS expansion into clinical settings, but up to now it has not yet been standardized. Several scientific studies have highlighted the prognostic role of NGS at diagnosis and during AML treatment.

A landmark study by Jongen-Lavrencic et al. [46] detected at least one mutation in 430 out of 482 (89.2%) patients enrolled. Mutations persisted in 51.4% of those patients during the first CR and were present at various VAF (range 0.02 to 47.0%). In the same analysis, the detection of persistent DTA mutations, often present in persons with age-related clonal hematopoiesis, was not correlated with increased relapse rate. Excluding DTA mutations in the analysis, the detection of NGS-MRD positivity was associated with a significantly higher relapse rate than no detection (55.4% vs. 31.9%; p < 0.001), as well as with lower relapse-free survival (RFS) (36.6% vs. 58.1%; *p* < 0.001) and OS (41.9% vs. 66.1%; *p* < 0.001). Multivariate analysis confirmed that the persistence of non-DTA mutations during CR conferred significant independent prognostic value with respect to the rates of relapse (*p* < 0.001), RFS (*p* = 0.001), and OS (*p* = 0.003). Moreover, a comparison of NGS-MRD vs. MFC-MRD showed that NGS had a significant additive prognostic value.

The aim of the study of Morita et al. [35] was to determinate whether the degree of mutation clearance at remission predicted the risk of relapse in 131 AML patients in morphological CR at day 30 after IIC. They analyzed the association between mutation clearance (MC) and event free survival (EFS) and OS cumulative incidence of relapse (CIR).

MC was evaluated based on VAF at CR with three thresholds: MC 2.5 as VAF of residual mutations < 2.5%; MC 1.0 as VAF of residual mutations < 1.0%; complete MC (CMC) as no residual mutations were detected.

As expected, significantly better OS and lower CIR was associate with MC 1.0 (2-year OS MC 1.0 vs. no-MC 1.0: 75% vs. 61%—*p* value = 0.04; 2-year CIR MC 1.0 vs. no-MC 1.0: 26% vs. 46%—*p* value = 0.03) and to CMC (2-year OS CMC vs. no-CMC: 77% vs. 60%—*p* value = 0.03; 2-year CIR: 24% vs. 46%—CMC vs. no-CMC; *p* value = 0.03). No significant difference in outcome by MC 2.5 was detected.

In multivariable analysis, patients with CMC had significantly better EFS (*p* = 0.008), OS (*p* = 0.04) and CIR (*p* < 0.001) than no-CMC patients. When DTA mutations were removed from the analysis, the prognostic associations were even stronger [35].

Klco et al. [56] performed an NGS-MRD analysis on day +30 after induction chemotherapy (IC); they confirmed that the detection of leukemia-associated mutations in patients in mCR was associated with an increased risk of relapse and reduced OS. Interestingly, in 13 patients, DNMT3A mutations persisted on day 30, and 12 of these 13 cases relapsed.

Onecha et al. evaluated—by NGS, MFC, and PCR—63 patients in CR after IC: 106 follow-up samples were analyzed according to level of mutations detected at diagnosis. The predictive value of MRD status by the three techniques was compared by survival analysis: NGS-MRD was positivity correlated with a lower disease-free survival (DFS) (*p* = 0.005) and OS (*p* < 0.001). In multivariate analysis, NGS-MRD positivity was an independent factor correlated to risk of death (*p* = 0.005) and of relapse (*p* = 0.012) [57].

These and other studies argue that persisting preleukemic mutations may be drivers of disease progression and predispose AML patients to relapse [34,35,46,54,55,56,57,58,59].

### 5.2. Allogeneic Hematopoietic Stem Cell Transplant Setting

With the improved safety of the procedure, disease recurrence continues to be the leading cause of treatment failure when alloHSCT is employed for AML. Appelbaum et al. compared the outcomes of approximately 1400 patients transplanted at their center from 1993 to 1997 with a similar cohort transplanted a decade later (from 2003 to 2007), and found that transplant-related mortality (TRM) had dropped by half over the decade [60]. Ten years later, the same authors published a similar study comparing the 2003–2007 cohort to the 2013–2017 cohort; again, a significant reduction in TRM, by more than an additional third, was evident [61]. The improvement was evident, both for myeloablative and for reduced intensity conditioning transplant, and was due to reductions in TRM (virtually for every complication related to transplant). Therefore, disease relapse continues to be the major limitation of transplant for AML patients; thus, prevention of relapse is the most relevant research topic in this field, and finding molecular predictors for relapse are urgently needed considering that 16% to 51% AML patients still experience recurrence of disease after transplant.

Different from other tumors, in particular solid tumors in acute leukemia, it has become part of routine clinical practice to collect bone marrow (BM) or peripheral blood (PB) samples from diagnosis to post-treatment follow-up and/or relapse. This allows the tracking of mutations longitudinally without additional invasive biopsy procedures. Taking advantage of the analysis of longitudinal sample collection, there has been great interest in developing assays to measure a trace amount of AML cells after conventional chemotherapy, before and after transplant.

#### 5.2.1. Pre-Transplant Mutational Status

Morphological CR (mCR) in AML represents a highly heterogeneous state, with a variable amount of residual leukemia burden that ranges from truly cured to imminent relapse [38,40,46,62,63,64,65]. It has been shown that MFC-MRD for AML patients in mCR at the pretransplant timepoint can predict post-transplant relapse and survival [63,64,66]. Given the genetic etiology of AML [23,67,68] and the success of monitoring by PCR-MRD in acute promyelocytic and NPM1-mutated AMLs [40,69], there is growing interest in the detection and quantification of AML-associated mutations to estimate residual disease burden using NGS, either for predicting risk of relapse after transplant or for modulating transplant strategy (for example, conditioning regimens or post-transplant maintaining therapy).

In several studies, the pre-transplant somatic mutation clearance has been associated with post-transplant outcomes in AML patients.

Getta et al. [70] analyzed, by NGS-MRD and MFC-MRD, 104 AML patients before transplant. They found that mutations in DTA and Janus Kinase 2 (JAK2) genes were less likely to be cleared than NPM1, isocitrate dehydrogenase (NADP (+))1 and 2 (IDH1/2), and FLT3-ITD. They revealed that the concordance rate of MRD using the two techniques was 71.0%, postulating that discordance could be explained both by DTA mutations not detected by MFC and by persistence of residual leukemia mutations below the established thresholds for mutation classification. Again, as expected, MFC-MRD and NGS-MRD positivity before transplant was associated with relapse risk (MFC-MRD: *p* value = 0.016; NGS-MRD: *p* value = 0.003) and survival (MFC-MRD: *p* value = 0.05; NGS-MRD: *p* value = 0.059) after alloHSCT. Moreover, residual disease detected concurrently by MFC and NGS conferred the highest relapse risk compared with discordant status or both negative status (overall, *p* = 0.008). They concluded that, although MFC is universally applicable, the NGS approach for MRD provides additional information concerning differential clearance of disease alleles and clonal architecture before transplant.

To determinate the association of somatic mutations with relapse risk after transplant, Luskin et al. [71] retrospectively studied the pre-transplant genetic profile sequencing of 112 AML patients, 96 (86%) of whom had at least one mutation.

Tumor Protein p53 (TP53), Wilms Tumor protein (WT1), and FLT3-ITD mutations were associated with an increased relapse risk (*p* value = 0.009, 0.02, and 0.07, respectively), while the DNMT3A mutation was associated with a lower risk of relapse (*p* value = 0.04). Comparison of pre- and post-transplant genetic profiles revealed clonal evolution in all relapsed patients, including acquisition of actionable mutations. They concluded that genetic profiling is useful for assessing the risk of relapse after transplant and may be helpful for identifying strategies to reduce the overall risk. Repeat genetic profiling at relapse may unveil acquired actionable mutations, confirming that clonal evolution is also present at post-transplant relapse.

As there is conflicting evidence on the prognostic relevance of persistent pre-leukemic clones in AML patients in CR, Rothenberg-Thurley et al. characterized—by NGS—paired pre-treatment and remission samples from 126 AML patients. Mutation persistence (40% of patients with more than one mutation) was most frequent in DNMT3A-positive patients (65% of patients with mutations at diagnosis), Serine and Arginine Rich Splicing Factor 2 (SRSF2) (64%), TET2 (55%), and ASXL1 (46%), and was significantly associated with older age (*p* < 0.0001). In multivariate analyses (adjusting for age, genetic risk, and transplant), NGS positivity, as expected, was associated with inferior RFS (HR 2.34; *p* = 0.0039) and OS (HR 2.14; *p* = 0.036). Patients with persistent mutations had a higher cumulative incidence of relapse before, but not after, transplant. The authors concluded that alloHSCT abrogated the increased relapse risk associated with NGS positivity, suggesting that NGS-MRD analysis may help in modulating post-remission treatment [72].

Considering that molecular MRD assessment is not established in approximately 60% of AML patients due to the lack of suitable markers for PCR, to overcome this limitation, Thol et al. established an error-corrected NGS-MRD approach that can be applied to any somatic mutation. They evaluated the clinical significance of this approach in 118 AML patients undergoing alloHSCT as in mCR. NGS at diagnosis identified a suitable mutation in 93% of the patients. Subsequently, NGS-MRD was performed in patients in CR (both from peripheral blood and bone marrow samples) before transplant; 12 patients were excluded from the final analysis as positive for DTA mutations. The analysis was performed in the remaining 96 patients, of which 45% were MRD-positive. In competing risk analysis, CIR was higher in MRD-positive than in MRD-negative patients (5-year CIR, 66% vs. 17%, *p* value =< 0.001,), whereas TRM was not significantly different. In multivariate analysis, MRD positivity was an independent negative predictor of CIR (*p* value < 0.001). They concluded that NGS-based MRD is widely applicable to AML patients, is highly predictive of relapse and survival, and may help to refine transplant and post-transplant management in AML patients.

Ahn et al. [73] evaluated treatment outcomes according to NGS-MRD in 124 AML patients with normal karyotype. By sequencing samples collected at diagnosis and at first mCR, they identified 361 mutations at diagnosis and tracked these at mCR. After excluding mutations associated with CHIP, 82 mutations were detected at CR in 50 patients (40.3%). In multivariate analysis, survival benefit was observed in favor of alloHSCT over conventional consolidation chemotherapy in the MRD-positive subgroup with respect to OS (*p* value = 0.003), RFS (*p* value = 0.015) and CIR (*p* value = 0.004), but not in the MRD-negative subgroup. The authors concluded that, for AML patients with normal karyotype achieving mCR with conventional induction chemotherapy, transplant is indicated when persistent genetic mutations were detected by NGS.

##### Pre-Transplant Mutational Status and Conditioning Regimen

Taking together the prognostic role of NGS-MDR analysis before transplant and the superiority of alloHSCT in respect of conventional consolidation chemotherapy in NGS-MRD-positive patients [73], the next challenge is to improve post-transplant outcome of NGS-MRD-positive patients.

Hourigan et al. [74] performed NGS-MRD analysis on preconditioning blood samples from patients enrolled in a phase III clinical trial and randomly assigned patients in mCR to myeloablative (MAC) ore reduced-intensity conditioning (RIC). NGS-MRD-negative was detected in 32% of MAC and 37% of RIC patients; a similar survival was detected in the two groups (3-year OS, 56% vs. 63%; *p* value = 0.96).

Meanwhile, in NGS-MRD-positive patients, relapse and survival was significantly different between the MAC and the RIC arm (3-year CI, 19% vs. 67%; *p* value < 0.001; 3-year OS, 61% vs. 43%; *p* value = 0.02, respectively). In multivariate analysis for NGS-positive patients, adjusting for disease risk and donor group, RIC was significantly associated with increased relapse risk (*p* value < 0.001), decreased RFS (*p* value < 0.001), and decreased OS (*p* value = 0.01) compared to MAC.

The evidence that MAC rather than RIC in AML patients with genomic evidence of MRD before alloHSCT can improve post-transplant outcome is also demonstrated in subsequent analysis by Hourigan et al., presented at the 2022 ASCO Annual Meeting. AML patients in first CR at transplant and with pre-conditioning blood samples available in the Center for International Blood and Marrow Transplant (CIBMTR) biobank were included in the study. Of the 457 patients with samples available, 448 had sufficient clinical data and DNA for analysis; NGS-MRD positivity was detected in 129 pre-transplant samples (29%). As expected, and already highlighted in previous analysis, NGS-MRD positivity prior to transplant was associated with a 3-year RFS of 36% compared with 56% in those who were NGS-MRD-negative (*p* value < 0.001). Moreover, NGS-MRD impact was modified by conditioning intensity; positive patients receiving RIC had the highest relapse rate (57% at 3 years), while those who tested positive but received MAC had a relapse risk of 35% at 3 years (*p* < 0.001). This evidence provides the foundation for future precision medicine approaches for reducing post-transplant relapse by a correct modulation of conditioning regimen and eventually post-transplant maintaining treatment [74].

#### 5.2.2. Post-Transplant Mutational Status

Most of NGS-MRD analyses are performed during induction and consolidation treatment with conventional chemotherapy. Less is known in the context of transplant, particularly in the post-transplant setting [75,76], where additional important considerations should be done regarding the optimal timing for performing NGS analysis and how to interpret the results and their clinical implications [77], in particular after RIC regimens where NGS-MRD may result persistently positive for months after transplant (mixed chimerism) [9,39]. Another aspect regards how to interpret the acquisition of newly detected mutations of undetermined significance whose origin could be the donor [78].

Up to know, few studies have looked at the use of NGS for post-transplant MRD in AML, and only Kim et al. have analyzed longitudinally the prognostic relevance of persistent allelic burden after alloHSCT in 104 transplanted patients [79]. Samples were collected at diagnosis and at pre- and post-transplant (day +21, +90, +180, and +365). NGS detected 256 mutations in 86.5% of patients at diagnosis; a stepwise clearance was evident after chemotherapy and transplant. In a subset of patients, mutations were still detectable pre- and post-transplant. Most post-transplant mutations were those detected at diagnosis.

Final analysis demonstrated that post-transplant allelic burdens (day + 21) in relapsed patients were higher than in non-relapsed ones, those mutations detected at diagnosis all expanded in post-transplant relapse, and that VAF > 0.2% post-alloHSCT is associated with an increased 3-year relapse risk (56.2% vs. 16.0%; *p* value < 0.001) and a worse 3-year OS (36.5% vs. 67.0%; *p* value = 0.006). Moreover, multivariate analysis confirmed that VAF > 0.2% post-transplant is an adverse prognostic factor for survival (*p* value = 0.003) and relapse (*p* value =< 0.001), independent of the European LeukemiaNET risk group.

Hamilton et al. [80] performed targeted NGS analysis on 112 AML patients who underwent alloHSCT; the most common mutations were TET2 (14.7%), FLT3 (12.9%), DNMT3A (12.1%), and Runt-Related Transcription Factor 1 (RUNX1) (7.8%).

TP53 and Enhancer of Zeste Homolog 2 (EZH2) mutations were associated with poor RFS (*p* value = 0.017 and =0.003, respectively). The dismal outcomes associated with a mutational status of TP53 is well known and has been consistently observed across studies. In this regard, it is important to highlight a recent study by Clarke et al., in which they disclosed two new high affinity pharmacological chaperones stabilizing the oncogenic p53 mutant Y220C in vitro. The study represents a significant advance for future clinical testing [81].

Hamilton et al. also identified several other mutations (TET2, SRSF2, and other) associated with higher non-relapse mortality. Considering that there are increasing data on the association of TET2 as an important regulator of inflammation [82,83], this may potentially explain its role in contributing to worse outcomes post-transplant. Furthermore, recent studies have demonstrated progression and worse prognosis in patient with chronic congestive heart failure harboring TET2 and DNMT3A mutations. It could be that these mutations may contribute to inferior survival post-alloHSCT by a similar mechanism [84,85].

Several studies have shown that mutations in the DTA genes did not discriminate relapse risk in AML patients in first remission after IIC [46,74,81,86,87]. As in the alloHSCT setting, it would be conceivable that the reappearance of recipient cells after transplant indicates relapse, and that DTA mutations could be prognostic after transplant [44]. Up to now, also in the post-transplant setting, the prognostic role of DTA mutations remains moot.

Heuser et al. [52] evaluated the prognostic role of DTA and non-DTA mutations post-transplant in 154 AML patients. At least one mutation was detected in 138 (90%) patients, which where retrospectively monitored by NGS on day + 90 and/or +180 post-transplant. The rate of MRD positivity was similar when DTA and non-DTA mutations were considered separately (17.6% vs. 19.8%). They found that DTA mutations had no prognostic impact on CIR, RFS, or OS. While non-DTA MRD positivity was an independent adverse predictor of CIR, RFS, and OS, but not of NRM. This prognostic effect resulted independently of different cutoffs of VAF. MRD-log reduction between diagnosis and post-alloHSCT assessment had no prognostic value.

Kim et al. [88] investigated the role of NGS-MRD in the setting of transplant with the purpose of clarifying the optimal timepoints and cutoff values and the role of DTA, CHIP, and different transplant strategies. They analyzed samples and clinical data from 132 patients and longitudinally tracked clonal changes before and after transplant. A total of 389 somatic mutations in 47 genes with a median of three mutations/patient were detected.

They observed a significant reduction in VAF from diagnosis to post-transplant as follows: initial samples: median VAF—34.39% (IQR: 10.8–45.87%); pre-transplant: median VAF—2.69% (IQR: 0.38–16.36%); post-transplant: median VAF—0.19% (IQR: 0.13–0.60%).

Looking at molecular pathways, chromatin/cohesion, DNA methylation, and RNA splicing had lower mutational clearance than conventional chemotherapy: they further cleared (>80%) with transplant procedure.

As expected, NGS-MRD-positive patients had a significantly greater risk of relapse than those without persistent mutations (pre-alloHSCT: 34.8% vs. 6.7%, *p* value =< 0.001; post-alloHSCT: 43.5% vs. 12.3%, *p* value =< 0.001), resulting in inferior survival.

To determine the optimal VAF threshold for predicting post-transplant relapse, they compared various cutoffs (0%, 0.2%, 1.0%, 2.0%, 2.5%, and 5.0%), finding that complete clearance of mutations (VAF 0%) was independently associated with CIR and survival in the multivariate analysis at each timepoint.

Moreover, in this study, isolated detectable mutations in genes associated with clonal hematopoiesis were also significant predictors of post-transplant relapse.

They also found that the optimal timepoint of NGS-MRD assessment depended on the conditioning intensity (pre-transplant for myeloablative conditioning and post-transplant for RIC). They concluded that NGS-MRD detection had a prognostic value, both pre- and post-transplant, regardless of the mutation type, depending on the conditioning intensity.

Of note, not all patients with NGS-MRD positivity post-transplant subsequently relapse, suggesting the possible role of an emergent graft-versus-leukemia effect in disease control. However, the detection of MRD positivity post-transplant requires urgent in-depth consideration for prompt interventions such as the use of donor lymphocyte infusions (DLI) and/or target therapy. This is because any interventions are more effective with a lower disease burden (molecular relapse prior to hematological relapse) in a setting of pre-emptive treatment.

Moreover, post-transplant pharmacological interventions may synergically act with a conditioning regimen, increasing cytotoxic effects and enabling sufficient time for the development of an alloreactive T and B cell response against leukemia. Moreover, it is now well known that some pharmacological agents may interact with the immune system of the donor, accelerating the development of a graft-versus-leukemia effect, as seen in patients treated with sorafenib post-transplant, where a metabolic re-programming of leukemia alloreactive T-cells has been demonstrated [89].

Up to know, it is still unclear whether maintenance strategies in a post-transplant setting really prevent relapse or only delays it, which relates to uncertainty over the duration with which these agents should be used. Meanwhile, it is clearer that such strategies could allow the development of an effective graft-versus-leukemia effect thanks to immune reconstitution (Figure 2).

## 6. Clonal Evolution Detected by Next-Generation Sequencing: From Diagnosis to Post-Transplant Relapse

It is well known that AML patients at relapse can present the same genetic lesion observed at diagnosis or can have higher clonal complexity through the acquisition of new mutations, or lose some mutation, or both.

From a pathophysiological point of view, the prevailing hypothesis is that a previously present minor subclone may exert a survival advantage under the selective pressure of chemotherapy [50].

To study the mechanisms of therapy resistance and disease progression, Greif et al. [90] compared the coding mutational patterns of matched diagnosis, remission, and relapse samples from 50 AML patients submitted to IIC. Both molecular devolution and evolution were widely detected at relapse, with a predominance of the latter. Moreover, they observed that alterations of epigenetic regulators (for example, KDM6A (Lysine Demethylase 6A) mutation associated to cytarabine resistance) were frequently gained at relapse; low KDM6A expression correlated with adverse clinical outcome.

This and other studies provide examples of clonal stability and clonal devolution all occurring either alone or in all possible combination at relapse [50,91,92].

Another pathophysiologic mechanism of clonal evolution during relapse is that the clone at relapse could be therapy-related and thus independent of the previous clone. In a landmark study, Ding et al. [49] analyzed (at diagnosis and at relapse) the mutational patterns of eight AML patients.

They found two major clonal evolution patterns during relapse: in one, the clone present at diagnosis acquired mutations and evolved into the relapsed one; in the other, a subclone of the founding clone survived during treatment, acquired other mutations, and expanded at relapse.

In any cases, chemotherapy failed to eradicate the founding clone. The analysis of relapse-specific clones versus primary clone mutations in all eight AML patients revealed an increase in transversions, maybe due to DNA damage caused by treatment.

All these studies provided insights into the AML clonal structure, revealing a clonal dynamic throughout the course of treatment and a refined risk stratification in ICC-treated patients, switching from a static risk classification (diagnosis) to a dynamic one (during treatment).

Given the distinct kinetic and curative mechanisms in alloHSCT, compared to conventional chemotherapy, it is plausible that different disease-related AML-specific mutations may be detected and targeted at post-transplant relapse.

In detail, chemotherapy could impart a selective pressure on AML clonal architecture of a relatively short duration, while the immunological effect of the donor immune system is prolonged overtime and should be overcome for relapse to occur [93]. Moreover, it could also be assumed that the donor immune system itself could modulate pressure on leukemia clonal architecture.

This poses additional challenges for highly sensitive methods of MRD assessment, as the persistence of molecular abnormality post-transplant does not always imply imminent relapse [9,93] (Figure 3).

Post-transplant administration of either cellular or pharmacological therapy represents a key approach to reducing disease relapse. Such therapies can augment the immunological effect of transplant (for example, by donor lymphocyte infusions and/or checkpoint inhibitors), or they can modulate disease relapse kinetics (for example by FLT3 inhibitors and/or hypomethylating agents). Obviously, rational employment of post-transplant treatment would be aided by a deeper understanding of actionable leukemia mutations in recurrent disease (Figure 2).

Kim et al. [93] presented an NGS-MRD analysis of a single case at 12 timepoints, from the initial diagnosis until a fourth relapse. The analysis tracked the hierarchy of genetic acquisition, and subsequently described the evolution of somatic variants to infer clonal structure.

New subclones appeared after the first mCR; the presence or absence of different subclones during remission and relapse implies differing drug responses among subclones. They concluded that NGS analysis-compared remission and relapse provides a comprehensive view of clonal structure and evolution.

Quek et al. [91] studied changes in clonal structure-compared mutational profiles in bone marrow samples at diagnosis and at relapse in 19 AML patients. In 13 patients, mutational profiles were altered at relapse, and in 9 patients, mutations at relapse were not presented at diagnosis. In 15 patients, additional samples, available pre-transplant, were analyzed; in two patients, mutations identified post-transplant but not at diagnosis were detectable immediately prior to transplant.

Kim et al. [88] analyzed clonal dynamics evolution after transplant in their series; bone marrow samples at relapse were available in 17 patients. Sixteen patients (94.0%) had, at post-transplant relapse, some or all of the same mutations detected at diagnosis. In three patients, longitudinal tracking revealed detectable mutations at 2 or 3 months before relapse. In addition, one patient with a Kirsten Rat Sarcoma virus (KRAS) mutation at initial diagnosis showed a DNMT3A mutation post-transplant, which was thought to be of donor origin. Furthermore, three clonal mutations of the FLT3, Neuroblastoma RAS (NRAS), and Protein Tyrosine Phosphatase (PTPN) genes had evolved at 29 months post-transplant.

In the analysis of Kim et al. [79], clonal evolution of allelic burden, detected at diagnosis and at post-transplant, showed a significant reduction of allelic burden, both after ICC and after transplant. Among the 256 mutations with detected VAF over 2% at initial diagnosis, only 42 mutations (16.4%) remained at a VAF over 2% pre-transplant, whereas 114 of the initial mutations were eradicated. Twenty-three out of 104 patients (22.1%) relapsed post-transplant, and samples at relapse were available for 20 patients; these samples were sequenced for tracing clonal evolution from diagnosis to relapse. VAFs of mutations detected in the initial clone and in the post-transplant relapse were comparable (28.2% vs. 28.4%). Among the 61 mutations detected from longitudinal monitoring in relapsed patients, 37 were stable (60.6%), 9 were cleared (14.8%), and 15 were acquired/selected at relapse (24.6%). At least one mutation shared between the initial clone and the post-transplant relapse clone was detected in 17 (85%) patients. In the remaining three patients, NGS analysis revealed that the post-transplant relapse clone shared at least one mutation with the initial leukemia clone in two patients. Altogether, in 19 out of 20 relapsed patients (95%), the clone at relapse shared at least one mutation with the clone at diagnosis.

## 7. Conclusions

The prognostic of most AML patients is poor due the persistence of leukemia initiating cells, even after intensive treatment and transplant. Despite its limitations, NGS represents a useful tool to investigate both DNA and RNA in the MRD setting to better understand the dynamics of genetic lesions (e.g., persistent or relapse-specific mutations) and to improve prognostic and disease monitoring capabilities. Several studies have demonstrated that NGS-based mutational profiling provides a deeper understanding of mutation dynamics throughout the course of AML and its clinical relevance in different settings compared to MFC.

Although DNA mutational assays provide information on biology and prognosis, bulk sequencing cannot distinguish which mutations occur in the same clone and resolve the clonal architecture, particularly with rare variants detected in remission, which can impair the ability to identify relapse-causing MRD. In this context, single-cell sequencing (SC-SEQ) reduces these limitations, allowing one to evaluate the clonal dynamics changes from diagnosis, through remission, to relapse [94]. At the same time, SC-SEQ characterizes and depicts changes in clonal architecture, providing insight into the clinical relevance of co-occurring clonal mutations. In fact, several published papers have suggested that SC-SEQ-based evaluation of MRD during CR may improve both the identification of AML patients at high risk for relapse and the precise monitoring of stem cell engraftment after transplantation [94,95]. Despite the sequencing costs, in the near future this approach could be routinely applied in clinical practice to outline tailored approaches to treat or to prevent relapse in AMLs (Figure 4).

MRD monitoring (by MFC, RQ-PCR, and NGS) and the assessment of clonal evolution (by NGS) from diagnosis to transplant should be made mandatory as soon as possible in order to more accurately identify patients who need to be submitted early for transplant and those who are candidates for pre-emptive or maintenance therapy after transplant. This approach required a continuous collaboration between AML teams (with the cell team), laboratory teams, and transplant teams from diagnosis to post-transplant follow-up (Figure 5).

## Figures and Tables

**Figure 1 biomedicines-11-00359-f001:**
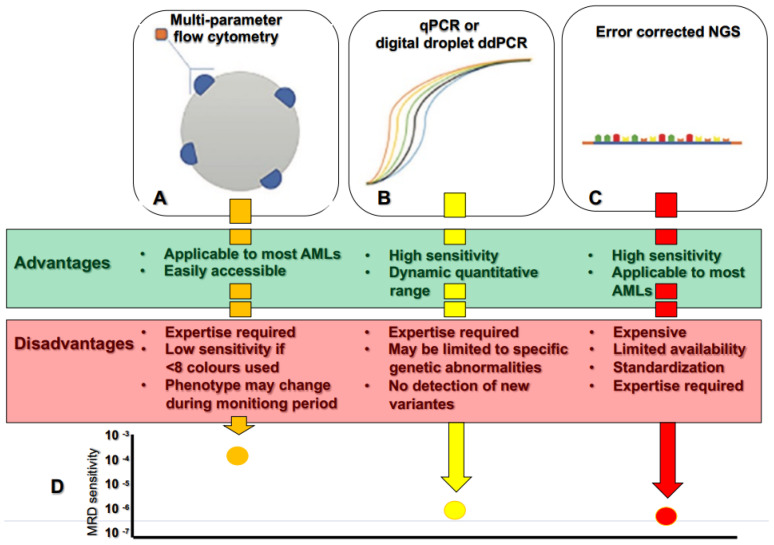
Summary of the advantages and disadvantages of methods of MRD assessment: (**A**)—Multi-parameter flow cytometry; (**B**)—Quantitative polymerase chain reaction (PCR) and droplet PCR; (**C**)—Next generation sequencing; (**D**)—The different sensitivity MRD assessments comparing the ability of each method to detect a single AML cell amongst normal haemopoietic cells. Sensitivities are as follows: Flow cytometry, 1 in 10,000; qPCR, 1 in 1,000,000; ddPCR, >1 in 1,000,000; NGS, 1 in 1,000,000.

**Figure 2 biomedicines-11-00359-f002:**
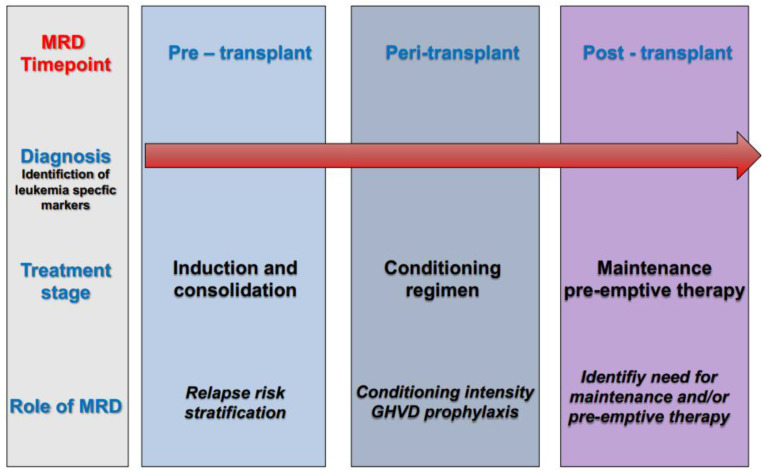
Role of measurable residual disease (MRD) at different timepoints of the treatment pathway in acute myeloid leukemia (AML).

**Figure 3 biomedicines-11-00359-f003:**
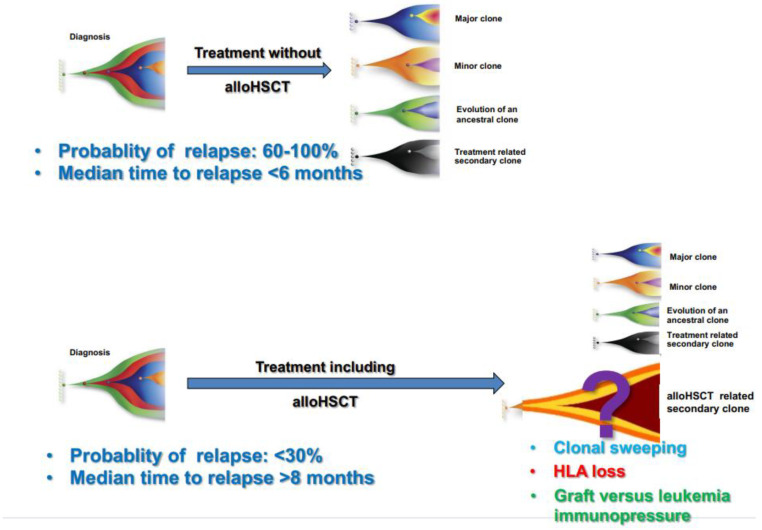
Clonal evolution and probability of relapse after conventional chemotherapy without transplant and after a program including transplant.

**Figure 4 biomedicines-11-00359-f004:**
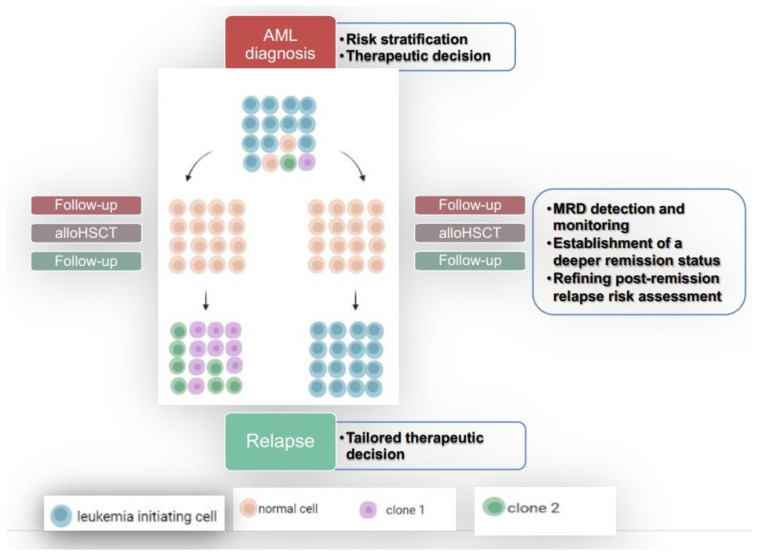
Role of single-cell sequencing from diagnosis to relapse.

**Figure 5 biomedicines-11-00359-f005:**
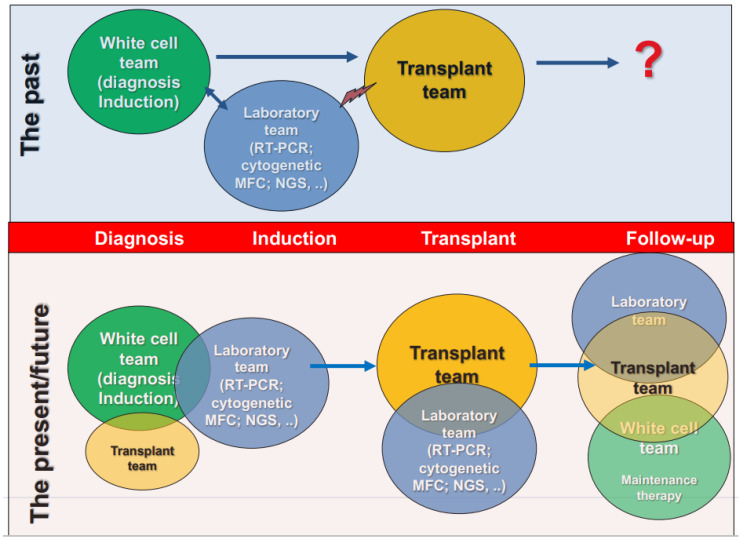
New approach for AML treatment: from diagnosis to post-transplant follow-up.

## Data Availability

Not applicable.

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
