# Peer review of "Measurable Residual Disease and Clonal Evolution in Acute Myeloid Leukemia from Diagnosis to Post-Transplant Follow-Up: The Role of Next-Generation Sequencing"

_biomedicines, 2023, doi:10.3390/biomedicines11020359_

Round 1

Reviewer 1 Report

Study:  MEASURABLE RESIDUAL DISEASE AND CLONAL EVOLUTION IN ACUTE MYELOID LEUKEMIA FROM
DIAGNOSIS TO POST TRANSPLANT FOLLOW-UP: ROLE OF NEXT-GENERATION SEQUENCING

Authors: Alessandra Sperotto, Maria Teresa Bochicchio, Giorgia Simonetti, Francesco Buccisano, Jacopo
Peccatori, Simona Piemontese, Elisabetta Calistri, Giulia Ciotti1, Elisabetta Pierdomenico,
Roberta De Marchi, Fabio Ciceri, Michele Gottardi

This review provides a general overview of the role of next-generation sequencing in optimize, and improve treatment outcome of patients suffering from acute myeloid leukemia. Here my comments
1.    Please remove all bulled lists scattered throughout the manuscript and try to describe the notions/statements in a more prosaic form
2.    Since AML being the main topic of the ms, a brief section describing the most important information of this tumor, such as epidemiology, survival, and current diagnostic, prognostic and therapeutic options should be included after the introduction.
3.    Please avoid the use of acronyms in the subhead titles. In addition, acronyms should always mentioned with their complete name when firstly described (is the case of “AMLs” or NGS in the introduction). The entire text should be carefully revised, accordingly. IN a similar way, gene should be mentioned with their complete name the first time being mentioned.
4.    “Unlike other disease that are characterized by a single somatic ……. making each patient genetically unique [9-13]. “  This reference should be included (PMID: 33958699)
5.    These panels cover between …….and cohesion-complex genes…), as entire genes or mutational hotspots” please include a reference
6.    “10^-4” it is unclear why the authors did not used the upper score
7.    A brief explanation of the factors/enzymes/genes mentioned should be included e. g., DNMT, TET and many others..
8.    In figure 1, there is no mention of the droplet PCR, which is more sensitive than conventional qPCR (PMID: 35741115)
1.    “Furthermore, recent studies demonstrated progression and worse …. inferior survival post alloHSCT by similar mechanism” A reference should be included PMID: 35339897 and PMID: 35296003
2.    Page 11, “reconstitution. (Figure 2)” please correct the typo
3.    “By a pathophysiological” should be “from a pathophysiological”
4.    “and other studies [86-88]” refs can be oved at the end of the sentence
5.    I suggest improving the contrast of figure 4 as being difficult to observe
6.    “figure 5” at the top of figure 5 can be removed. In addition, the figure seems to be with low quality, please improve the quality.

Author Response

PLEASE, SEE THE ATTACHMENT

THANK YOU

Reviewer 2 Report

Alessandra Sperotto and colleagues present a quality and well-written review manuscript focused on measurable residual disease and clonal evolution in acute myeloid leukemia from diagnosis to post transplant follow-up with the emphasis role of next-generation sequencing.

Authors provide an overview of the role of NGS in monitoring both MRD and clonal evolution in AML patients during the entire course of the disease, with particular regard to transplant phase.

Authors argue that one of the most important advances in this setting was the introduction of NGS, which enables reliable detection of patient-specific mutations covering complete gene assessment at diagnosis, complete remission and relapse. Despite the relevance of NGS at diagnosis is now ascertained, its role in other phase of the disease – during treatment: at relapse, before and after transplant – is less clear. Therefore they attempted to provide an overview of clonal evolution of AMLs, monitoring by NGS, during the entire course of the disease, including the alloHSCT setting.

Authors suggest that DNA mutational assays provide information on biology and prognosis, bulk sequencing cannot distinguish which mutations occur in the same clone and resolve the clonal architecture, particularly with rare variants detected in remission, which can impair the ability to identify relapse causing MRD. In this context, single cell sequencing reduces these limitations allowing to evaluate the clonal dynamics changes from diagnosis to remission to relapse. At the same time, SC-SEQ characterize and depict changes in clonal architecture providing insight into the clinical relevance of co-occurring clonal mutations. In fact, several published papers, suggested that SC-SEQ – based evaluation of MRD during CR may improve both the identification of AML patients at high risk for relapse and the precise monitoring of stem cell engraftment after transplantation. Despite the sequencing costs, in the next future, this approach could be routinely applicated in clinical practice to outline tailored approaches to treat or to prevent relapse in AMLs.

Finally, authors conclude that the MRD monitoring and the assessment of clonal evolution from diagnosis to transplant, should be mandatory as soon as possible, in order to identify more and more accurately patients to be submitted early to transplant and those to be candidate to pre- emptive or maintenance therapy after transplant. 

Overall, the manuscript is valuable for the scientific community and should be accepted for publication after edits are made.

===========================

Other comments:

1) Please check for typos throughout the manuscript.

2) Page 8 and page 10. With regards to TP53 mutations – authors are kindly encouraged to cite the following article that describes development of novel therapeutics targeting mutant p53. DOI: 10.1021/acsptsci.2c00164

Author Response

PLEASE, SEE THE ATTACHMENT

THANK YOU
